# Empowering Youth Through Evidence: Applying Outcome Harvesting to Evaluate Sexual Reproductive Health and Rights (SRHR) Advocacy in Ethiopia

**DOI:** 10.3390/ijerph22111659

**Published:** 2025-11-01

**Authors:** Mihret Ayele, Makida Birhan, Sintayehu Abebe, Mesfin Ayeta, Dessie Kassa, Muluken Dessalegn Muluneh, Wendemagegn Enbiale

**Affiliations:** 1Amref Health Africa, Addis Ababa P.O. Box 20855, Ethiopia; makida.berhan@amref.org (M.B.); sintayehu.abebe@amref.org (S.A.); mesfin.ayeta@amref.org (M.A.); dessie.kassa@amref.org (D.K.); muluken.desalegne@amref.org (M.D.M.); 2Department of Dermatovenereology, College of Medicine and Health Sciences, Bahir Dar University, Bahir Dar P.O. Box 78, Ethiopia; wendaab@gmail.com; 3Collaborative Research and Training Center for Neglected Tropical Diseases, Arba Minch University, Arba Minch P.O. Box 21, Ethiopia

**Keywords:** youth empowerment, outcome harvesting, SRHR, adolescents, Ethiopia, participatory evaluation

## Abstract

Adolescent sexual and reproductive health and rights (SRHR) challenges, including gender inequality, child marriage, female genital mutilation/cutting and sexual and gender-based violence remain pervasive globally, particularly in Ethiopia. The Power to You(th) programme was designed to address these issues by centering youth voices and fostering transformative community change. This evaluation applied Outcome Harvesting, drawing on mixed quantitative and qualitative data, and employed a participatory approach that engages youth and stakeholders in identifying and verifying programme outcomes between 2021 and 2023. The findings revealed substantial improvements in youth participation in decision-making platforms, advocacy capacity, and awareness of SRH rights. Community attitudes shifted positively, particularly among religious and traditional leaders, who increasingly advocate against harmful practices. Youth-led networks emerged as powerful agents of change, contributing to policy shifts such as increased youth representation in health governance bodies. The evaluation also captured unexpected yet influential changes in community dynamics and institutional responsiveness. These findings highlight the value of participatory approaches in capturing complex social transformations and underscore the role of youth as active agents in reshaping SRHR outcomes. Outcome Harvesting proved effective in documenting both intended and emergent changes, offering valuable insights for scaling inclusive, youth-driven interventions.

## 1. Introduction

Globally, adolescent sexual and reproductive health and rights (SRHR) remain a pressing public health and human rights concern, particularly for adolescent girls and young women (AGYW) in low- and middle-income countries. AGYW frequently face heightened vulnerability due to entrenched gender inequalities, harmful traditional practices such as child marriage and female genital mutilation/cutting (FGM/C), and pervasive sexual and gender-based violence (SGBV) [1,2]. These practices severely compromise young women’s health, educational opportunities, and long-term development prospects [3,4]. In Ethiopia, despite notable progress over the past decade, significant challenges persist, including persistently high rates of child marriage and FGM/C, underscoring the urgent need for targeted, effective interventions [5].

Previous interventions have largely relied on conventional strategies, such as mass awareness campaigns, school-based education programs, and community mobilization efforts, often incorporating advocacy aimed at influencing behavior and policy change [6,7]. Recent program evaluations, including those from initiatives such as the “Get Up, Speak Out (GUSO)” and the UNICEF-UNFPA Joint Programme on FGM/C, have documented meaningful progress in raising awareness and shifting attitudes [6,8]. However, limitations are evident in the existing literature. Many evaluations often prioritize predetermined, quantitative indicators, neglecting nuanced qualitative changes, unintended outcomes, and context-specific shifts in community norms and individual behaviors [9,10]. Moreover, a common critique of youth-focused SRHR interventions is the insufficient genuine involvement of youth themselves in program design, implementation, and evaluation, which can undermine sustainability and effectiveness [11,12,13]. Conventional monitoring and evaluation frameworks, such as logical frameworks or indicator-based assessments, are useful for tracking outputs but often miss complex, non-linear changes in social norms, power relations, and youth agency. These approaches are less suited for adaptive advocacy contexts, where outcomes emerge unpredictably through negotiation among multiple stakeholders [14].

Outcome Harvesting (OH), by contrast, is a complexity-aware evaluation approach that does not rely on predefined indicators. Instead, it retrospectively and prospectively identifies observable changes in behaviors, relationships, policies, or practices among social actors, making it particularly well-suited to dynamic advocacy programs like Power to You(th). By systematically substantiating outcomes with multiple stakeholders, OH enhances both credibility and comprehensiveness, capturing a richer picture of social transformation [15].

As a relatively novel participatory method, OH has gained traction for its ability to systematically document both intended and unintended outcomes in complex contexts [16,17,18]. By involving stakeholders actively in outcome identification and validation, OH strengthens local ownership and the rigor of evaluation of findings [11,12,13,14,15,16,17]. Recent global application such as the “Partnership to Inspire, Transform and Connect the HIV response (PITCH)” initiative and ActionAid’s “Africa We Want” project, have shown its effectiveness in capturing comprehensive, context-sensitive outcomes, including policy advocacy and behavioral change among targeted populations [19,20].

Despite the proven potential of OH, its application in youth-focused SRHR programs, particularly within Ethiopia, remains limited. Most existing evaluations in Ethiopia have predominantly employed conventional methodologies, potentially missing critical emergent outcomes central to understanding how youth empowerment unfolds in practice [19]. The literature thus points toa gap in robust participatory and complexity-aware evidence capable of capturing youth-led and youth-centered outcomes within SRHR interventions [7,12,21].

In response, the Power to You(th) (PtY) programme in Ethiopia, funded by the Dutch Ministry of Foreign Affairs, represents an important innovation by combining meaningful youth participation, a gender-transformative approach, and a strong commitment to Southern Leadership principles. The programme uniquely integrates OH to comprehensively capture empowerment outcomes beyond predefined indicators [10,22]. Through OH, PtY systematically document emergent, youth-driven changes in real-time, offering deeper insights into how young people contribute to shifting norms, policies, and practices regarding SRHR in Ethiopia.

This manuscript presents findings from the mid-term evaluation of the PtY programme in Ethiopia, focusing on harvested outcomes. The objectives are to assess the programme’s contribution to enhancing meaningful youth participation, improving SRHR awareness and advocacy capacity, and challenging harmful traditional practices among Ethiopian youth.

## 2. Materials and Methods

### 2.1. Study Design and Approach

This evaluation employed a participatory, mixed-methods design, with Outcome Harvesting (OH) serving as the primary methodology to assess the implementation and early outcomes of the Power to You(th) (PtY) programme in Ethiopia.

Given the programme’s complexity—addressing youth empowerment, civic engagement, and efforts to eliminate harmful traditional practices (HTPs)—OH was selected for its flexibility to capture both intended and unintended outcomes. Although typically applied retrospectively, in this study OH was applied both retrospectively and prospectively. This dual application allowed the evaluation to not only document past changes but also anticipate and adapt to emerging outcomes, thereby supporting continuous learning and reflection during programme implementation. The OH process followed six iterative steps [9] (Table 1):

To ensure rigor and credibility, OH was guided by nine foundational principles, including participatory engagement, plausibility of contribution, and harvesting verifiable social change outcomes.

Outcome harvesting was initiated at the start of the PTY program, with an outcome logbook systematically capturing all outcomes—defined as observable and significant changes in a social actor’s behavior, relationships, activities, policies, or practices. Midway through implementation, an Outcome Harvesting workshop was conducted with all stakeholders and implementers who were familiar with the project’s Theory of Change (TOC) and had reported outcomes. During the workshop, the team carried out a thematic analysis to categorize outcomes along TOC pathway. A total of 14 key outcomes were then prioritized through a participatory process with programme implementers and stakeholders. Outcomes were selected based on their relevance to the Theory of Change (TOC) pathways of youth empowerment, gender norm transformation, and SRHR advocacy. In some cases, outcomes that were not explicitly anticipated in the TOC but that directly advanced the overarching programme goal were also included. Prioritization was further informed by evidence of significant programme contribution and stakeholder perception of the outcome’s importance (e.g., influencing policy, shifting community norms, strengthening youth participation). This ensured that both expected and emergent, yet substantively important, outcomes were captured and analyzed.

This process was followed by a contribution analysis, which involved listing all contributors, determining their level of contribution, and situating each outcome within the programme’s overarching goal. For every outcome, the team articulated a clear contribution claim linking PtY activities to observed changes through specific mechanisms of influence. These claims were examined in participatory workshops where implementers and stakeholders collectively mapped the broader “causal package.” This mapping acknowledged that outcomes rarely result from a single factor, positioning PtY interventions (e.g., trainings, advocacy dialogues, youth club strengthening) alongside the efforts of other actors such as government initiatives, CSO partnerships, and contextual drivers. The emphasis was on how PtY created enabling environments or directly influenced actors, while recognizing that the realization of change ultimately rested with those actors themselves.

To test the plausibility of these contribution claims, evidence was gathered beyond narrative outcome statements. Programme documents, meeting records, policy circulars, and service statistics were reviewed, while follow-up interviews with youth champions, community leaders, and government officials provided testimonies about the timing and sequence of changes. Contribution was thus understood not as sole attribution to PtY, but as the plausible demonstration that PtY inputs significantly enabled or influenced observed changes.

### 2.2. Complementary Methods

To enrich and validate the OH findings, additional qualitative and quantitative methods were employed:Key Informant Interviews (KIIs): Conducted with youth champions, CSO representatives, government officials, and societal actors to gain deeper insight into context-specific changes and perceptions of programme effectiveness.Appreciative Inquiry: Used during workshops to identify success stories, strengths, and enabling factors that contributed to positive change.Advocacy Maturation Tool: Applied to assess the maturity and effectiveness of advocacy strategies led by youth and CSO actors.Desk Review: Programme documents, progress reports, and monitoring data were analyzed to provide background and support outcome verification.Descriptive Statistics: Quantitative data from programme monitoring tools were summarized to complement the qualitative findings and track programme reach.

These methods were strategically integrated to triangulate findings, enhance understanding of the change processes, and validate the programme’s contribution to observed outcomes.

### 2.3. Study Setting and Fieldwork

Data were collected from two clusters in the Amhara Region (North Shoa and Bahir Dar) and one site in the Afar Region. Three field teams were established for simultaneous data collection. Two teams operated in Amhara while the third focused on Afar. Field personnel were selected based on prior experience in qualitative research and deep familiarity with the local context.

### 2.4. Sampling for Verification

A purposive sampling strategy was used to select respondents with direct knowledge or experience of programme implementation and outcomes. Participants were identified through OH workshops and a substantiation plan developed with programme partners. In total, 51 interviews were conducted across categories: 30 substantiation interviews (25 from Amhara, 5 from Afar), 10 youth champions (8 from Amhara, 2 from Afar), and 11 key informants comprising CSO representatives, societal actors, and state officials (8 in Amhara, 3 in Afar). Additionally, 12 programme implementers participated in the OH workshop, and 13 stakeholders were engaged in the advocacy maturation assessment. This purposive approach ensured broad representation across youth, civil society, government, and community actors, thereby strengthening the methodological rigor and credibility of the findings

### 2.5. Data Analysis

Qualitative data, including interview transcripts, workshop discussions, and outcome statements, were thematically coded using NVivo version 15 software. Codes were structured around the PtY programme’s ToC, with key domains such as youth agency and leadership, gender norm transformation, civic participation and advocacy, government accountability and responsiveness.

Thematic analysis followed a two-stage process. In the first stage, inductive coding allowed themes to emerge directly from the data, ensuring that unanticipated or unintended outcomes were not overlooked. In the second stage, deductive coding mapped emergent themes onto predefined ToC domains, donor reporting requirements, and strategic priorities. This dual approach maintained fidelity to the data while ensuring alignment with programme objectives. For example, unexpected youth-led advocacy efforts or new religious leader engagement were mapped alongside existing ToC pathways, ensuring both emergent and anticipated outcomes were systematically captured.

Contribution analysis was applied to assess the plausibility of PtY’s role in each outcome. Quantitative data were analyzed using descriptive statistics including frequencies and percentages and to complement qualitative results and contextualize programme reach and outputs.

## 3. Results

### 3.1. Outcomes Verified

Through the processes outlined in the methodology, a total of 14 outcomes were systematically prioritized and substantiated for this evaluation. These outcomes, all fully confirmed, capture significant changes ranging from youth-led advocacy and prevention of harmful traditional practices to institutional commitments such as the establishment of free SRHR services. Table 2 summarizes the final outcome distribution, which includes youth successfully lobbying for the reinstatement of youth-friendly services, preventing FGM and child marriage, resisting gender-based violence, mobilizing CSO networks, and influencing both religious leaders and local governments.

To facilitate coherent interpretation, the results are presented across three core domains of the programme’s Theory of Change: (1) Youth Agency and Advocacy, (2) Changing Social Norms, and (3) Institutional and Policy Change. Each thematic area integrates Outcome Harvesting evidence with insights from Appreciative Inquiry youth stories, providing both outcome-level confirmation and reflections on the enabling factors that made change possible. Finally, findings from the Advocacy Maturation Assessment are presented in a separate subsection to highlight the evolving advocacy capacity of youth and CSOs.

### 3.2. Youth Agency and Advocacy

The PtY programme significantly strengthened youth agency and advocacy capacities across the intervention sites. Outcome Harvesting documented several cases where youth-led actions directly influenced service provision and decision-making processes. One illustrative example was in Shewa Robit, where members of the local youth club successfully lobbied woreda officials to reinstate Youth-Friendly Services (YFS) that had previously been repurposed for other uses.

A youth club leader explained, “We started to demand accountability after PtY. The youth-friendly service was previously used for other purposes”.

Substantiation interviews with health officials confirmed that this change occurred after sustained youth advocacy efforts, and monitoring data showed increased service utilization following the reactivation.

Youth also demonstrated agency in challenging harmful traditional practices through individual and collective action. For instance, a male youth advocate in Kewot Woreda prevented his sister’s circumcision explaining, “I used the knowledge from PtY to explain the health effects. My mother was convinced and cancelled it.”

Similarly, adolescent girls in Yinesa Kebele organized through their school’s girls’ club to resist sexual harassment. “Our future is in our hands and our priority is education”, one girl reported. These actions not only protected individuals but also signaled broader shifts in peer solidarity and youth voice within communities.

Findings from Appreciative Inquiry youth stories reinforced this evidence, showing that champions attributed their ability to engage decision-makers to the systematic chain of trainings they received. As one participant put it, “The continued trainings helped us to strengthen our voice, role, and position in society”. These reflections highlight the mechanisms by which PtY investments translated into youth confidence and advocacy effectiveness.

### 3.3. Changing Social Norms

The PtY programme contributed to meaningful shifts in community attitudes and practices that employ harmful traditional practices (HTPs). In Kewot Woreda, a former FGM practitioner renounced the practice and began advocating against it, stating, “After PtY training, I changed my perception completely”. Substantiation interviews confirmed her influence in encouraging families to abandon FGM.

Community leadership was also mobilized against child marriage. In Yemikat Kebele, a local leader led the cancellation of three planned child marriages after forming a harmful practice eradication committee, “We cancelled three child marriages with community support after PtY training changed my view”.

Religious leaders in Afar amplified this change by publicly endorsing gender equality and youth rights. One senior leader declared, “We have been working with youth representatives to tell the community that youth have the right to education and to decide for themselves”.

Appreciative Inquiry stories echoed these transformations, as youth champions emphasized how PtY equipped them to speak publicly on sensitive issues such as SGBV and unintended pregnancy. Participants described these moments as “turning points” in their communities, enabled by PtY’s consistent mentoring and safe spaces for dialogue. One young woman explained, “PtY made us realize that harmful practices are not our destiny—we have the right to change them”. This combination of Outcome Harvesting and Appreciative Inquiry evidence underscores how PtY’s strategies not only changed individual behavior but also facilitated broader shifts in community norms.

### 3.4. Institutional and Policy Change

The PtY programme also catalyzed institutional and policy shifts that increased responsiveness to youth PtY also catalyzed institutional and policy shifts that expanded access to SRHR services. In Asayita, local government established a one-stop centre providing free SRHR and legal services. A health office representative noted, “We established the centre to provide counselling, treatment, and legal services related to SRHR”. Youth inclusion in governance also expanded, with councils in Kewot and Bahir Dar Zuria formally adding youth representatives. Monitoring data confirmed youth participation in 29 health centre governing boards in Amhara.

Partnerships with religious and school leaders further institutionalized advocacy. A school principal described his engagement: “Since 2022, I have been advocating PtY’s programme everywhere. We provided life-skills and SRH training for more than 200 students”.

Appreciative Inquiry stories provided complementary insights into how these institutional shifts were experienced by young people. Youth champions emphasized that being invited into formal spaces of governance reflected a new recognition of their role, which they saw as evidence of their advocacy being institutionalized rather than tokenistic. One youth participant reflected, “For the first time, I felt that our voices were not only heard but also respected at the decision-making table”. Such reflections highlight how PtY’s trainings and mentorship prepared youth to navigate governance spaces, while advocacy outcomes confirmed that institutions were increasingly responsive to youth voices.

### 3.5. Advocacy Maturation Assessment

In addition to Outcome Harvesting and Appreciative Inquiry, the evaluation applied the Advocacy Maturation Tool (AMT) to assess the capacity of youth groups and CSOs to design and sustain effective advocacy strategies. The assessment, conducted with 14 participants from YNCD, Hiwot Ethiopia, and Amref Health Africa, found that progress was uneven but generally clustered around the consolidation stage (level 3).

Youth meaningful engagement by CSOs and improvements in lobbying and advocacy capacity were both rated at the consolidation stage, suggesting that youth and partner organizations had developed consistent practices for influencing decision-makers. Other areas, including the use of a gender and social inclusion lens, gender-transformative approaches, and accountability mechanisms, were rated slightly lower at 2.5, reflecting progress between emerging and consolidation stages. The overall average score was 2.7, underscoring that while youth and CSO advocacy capacity is strengthening, additional support is required to embed inclusive and accountable practices more systematically.

These results complement the Outcome Harvesting findings by showing that the programme not only achieved visible advocacy outcomes but also helped build the organizational and strategic capacities of youth and CSOs to sustain advocacy work in the longer term.

## 4. Discussion

Using Outcome Harvesting (OH), our evaluation identified significant outcomes achieved by the Power to You(th) (PtY) youth empowerment interventions in Ethiopia. These outcomes align closely with the program’s objectives of enhancing meaningful youth participation, increasing sexual and reproductive health and rights (SRHR) awareness, and addressing harmful traditional practices. The programme supported establishing and strengthening 40 youth clubs across 14 districts in Ethiopia, significantly improving life skills, SRHR knowledge, and advocacy capabilities among youth. Youth networks formed in seven districts to amplify their advocacy efforts, leading to greater participation and voice in community decision-making forums. The Mid-Term Evaluation (MTE) further revealed that youth knowledge of harmful traditional practices, sexual and gender-based violence (SGBV), and unintended pregnancies markedly increased. Youth secured representation on 29 health center governing boards in the Amhara region, integrating their perspectives directly into local governance. Additionally, youth-led dialogues engaged 60 religious leaders in the Afar region, fostering open discussions and challenging misconceptions surrounding practices such as female genital mutilation/cutting (FGM/C) and child marriage. Collectively, these outcomes indicate substantial progress towards the programme’s objectives.

This study’s use of OH in evaluating youth empowerment outcomes in SRHR contexts is innovative in Ethiopia. OH enabled the capture of unanticipated outcomes, such as the emergence of youth-led district networks and involvement of religious leaders, which traditional evaluation methods might overlook [9,10]. This evaluation demonstrated a groundbreaking shift toward youth inclusion in institutional and community decision-making spaces a historically rare phenomenon in Ethiopia [4,7]. By involving youth directly in governance structures and facilitating dialogues with community gatekeepers, the program successfully navigated deep-rooted cultural and social norms, particularly in conservative communities such as those in the Afar region [13]. These findings reinforce the importance of participatory evaluation methods in accurately capturing complex, non-linear social changes in youth empowerment programs.

The findings align with and expand existing literature on youth empowerment and participatory evaluation methods. Outcome Harvesting has increasingly been recognized as an effective approach to documenting outcomes in complex developmental contexts, particularly where traditional linear models are inadequate [9,10,11]. Similar to previous studies, this method’s participatory nature facilitated greater stakeholder engagement and ownership of results [10,12].

Comparable programmes such as Get Up, Speak Out (GUSO) and the PITCH initiative have documented significant increases in youth SRHR awareness and engagement. However, the Outcome Harvesting approach used in this evaluation provided deeper qualitative insights into the processes behind these changes, particularly in capturing how youth-led advocacy translated into concrete policy actions [22]. Our findings thus complement existing quantitative data by elucidating underlying mechanisms of change in youth empowerment interventions.

One particularly unexpected but significant outcome was that some youth resisted parental efforts to delay marriage. While parental discouragement of early marriage is generally celebrated, these cases reveal a more complex reality: young people asserted their agency by rejecting parental control, even when it was framed as protective. This underscores the non-linear and sometimes paradoxical nature of empowerment, where youth agency may not align neatly with adult or community expectations. OH was particularly well-suited to document such nuanced dynamics, which would likely have been missed through conventional indicator-based evaluations.

In Ethiopia, previous research highlighted limited youth participation in governance and policy-making roles due to entrenched cultural and structural barriers [4,7]. Our results demonstrate meaningful departures from these limitations, presenting clear cases of youth participation in governance and advocacy actions that directly influenced local policies and practices. Moreover, the study underscores successful strategies for overcoming barriers by engaging influential community actors, consistent with broader advocacy literature [7].

A key strength of this study lies in its participatory design, which actively engaged youth, CSOs, community leaders, and government actors in identifying and substantiating outcomes. This inclusivity enhanced credibility and strengthened local ownership of findings. Another strength is methodological triangulation: OH was complemented with Appreciative Inquiry stories, the Advocacy Maturation Tool, and programme monitoring data. Together, these methods captured both intended and unintended outcomes, illuminated the mechanisms driving change, and assessed the evolving advocacy capacity of youth and CSOs. This integration increased the robustness of the evidence and provided a comprehensive understanding of how advocacy interventions function in complex social settings.

Several limitations should be noted. First, while contribution analysis strengthened the plausibility of PtY’s role, OH cannot fully disentangle programme contributions from broader contextual influences, creating some risk of over-attribution. Second, outcome prioritization may have been subject to selection bias, as more articulate or accessible participants could have shaped which outcomes were emphasized. Third, the largely qualitative evidence base limited the ability to quantify the magnitude of change. Finally, the relatively short observation period constrained the assessment of long-term sustainability, an issue also noted in similar evaluations in Ethiopia.

## 5. Conclusions

The Outcome Harvesting evaluation demonstrated that youth empowerment interventions substantially enhance adolescent SRHR outcomes in Ethiopia. Participatory evaluation successfully captured complex, multi-level changes and emphasized the critical role of youth participation in achieving lasting social transformation. These findings highlight the importance of institutionalizing youth involvement in governance and policy processes and provide evidence for scaling youth-led advocacy models to strengthen SRHR programming. Future efforts should assess the long-term sustainability of these outcomes, expand mixed-methods approaches to capture broader community-level impacts, and explore intergenerational advocacy dynamics to deepen our understanding of effective empowerment strategies.

## Figures and Tables

**Table 1 ijerph-22-01659-t001:** The six iterative steps of an OH and its descriptions.

Steps	Descriptions
Design the Outcome Harvesting Method	During this stage the key questions that the OH is trying to answer are agreed. Agreement is also made on how evidence will be gathered and by whom in order to answer the key questions.
Review documentation	Potential outcome statements or descriptors are identified that include any changes observed in the social actors and how the intervention has influenced these changes.
Engage with informants	The harvester engages in discussions with those best placed to pass on knowledge about how outcomes have been achieved and who contributed to them.
Substantiate	Claims are substantiated by talking to external sources to ensure accuracy.
Analyse and interpret	Outcome statements are organised and the evidence gathered used to try and answer the initial evaluation questions
Support use of findings	Harvesters identify points for further discussion and ways in which the findings can be used in the future.

**Table 2 ijerph-22-01659-t002:** Final outcome distribution of the Outcome Harvesting study to Evaluate Sexual Reproductive Health and Rights (SRHR) Advocacy in Ethiopia.

S. No	Outcome Statement	Validation Status
1	In December 2022, Shewa Robit Youth Club members lobbied and advocated to the local government to make YFs functional to support with material	Fully Confirmed
2	In December 2022, Zerihun Tensitu, who is a member of Biruh Tesfa Youth Club, who lives in Kewote Woreda, Yelen kebele, managed to protect his younger sister from circumcision after strongly challenging his parents	Fully Confirmed
3	In September 2022, Tesfa Tewlid, PLWHIV/AIDS Biruh Tesfa PWD, Enat Charity and Ataye Elders association supported youth to get office space and exercise MYPE. They have also engaged decision-making bodies, advocating for youth rights and development in three target woredas (Kewet, Efrata Gedem and Ensaro), as well as Ataye Town Administration	Fully Confirmed
4	Mr. Teferi Negatu a former FGM cutter in Kewot Woreda stopped practicing FGM on November 2022	Fully Confirmed
5	Kewot district council decided to engage the young people in the district council at March, 2022 for the first time.	Fully Confirmed
6	Yeshambel Gebrie (his name was adjusted from Yeshambel Mogest to Yeshambel Gebrie and his place of residence was found to be Lata not Wugeta Kebele) a 28-year-old youth witness of child marriage from Bahirdar Zuria (Lata), became an advocate and exemplary to stop child marriage on 2022	Fully Confirmed
7	Hanna Tadesse, a 15-year-old adolescent girl in Yinesa Kebele of Bahrdar Zuria Woreda, stood firm and said no to violence/sexual harassment by a man claiming he loves her. The school girls’ club members stood in solidarity with Hanna and decided that their future is in their hands and their priority is their education.	Fully Confirmed
8	In July 2022, a Network of 20+ CSOs called NAGWA based in Bahir Dar City decided to start mainstreaming and advocating for key PTY issues	Fully Confirmed
9	On 6 June 2022, Estibel (Community Leader) led the cancelation of three child marriages at Yemiket Kebele, Woreda.	Fully Confirmed
10	In August 2021, Mr. Manaye, director of Debresina Junior Secondary School in Yinesa Kebele, Bahir Dar Zuriya Woreda, become an advocate of PtY issues and started to influence youths in the Kebele to take part in awareness raising	Fully Confirmed
11	On December 2021, the woreda heads from the sectors included youth representatives in the woreda-level administration committee in Bahir Dar and Bahirdar Zuria woreda	Fully Confirmed
12	In January–December 2022, SISD (local CSO in Asayita) mainstreamed Key Pty issues in their regular activities in Asayta and other woredas.	Fully Confirmed
13	In July 2021, the Afar Region Islamic Council Vice President and Afar Region Sheria Court President became Ambassadors of the PtY project.	Fully Confirmed
14	The local government in Asayita for the first time agreed to make the service at one stop center free and to also allocate attorneys to provide legal service on 24th December 2021	Fully Confirmed

## Data Availability

The data presented in this study are available on request from the corresponding author.

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
