# Peer review of "Empowering Youth Through Evidence: Applying Outcome Harvesting to Evaluate Sexual Reproductive Health and Rights (SRHR) Advocacy in Ethiopia"

_ijerph, 2025, doi:10.3390/ijerph22111659_

Round 1

Reviewer 1 Report

Comments and Suggestions for Authors

This is a well-written evaluation study using the OH framework to conduct a midterm evaluation of the Power to You(th) (PtY) programme in Ethiopia. Here are some minor comments for the authors:

  1. Please mention in the abstract that this is an evaluation study and which types of methods were used (qualitative/quantitative) to make it clear what type of study this is.
  2. The introduction was well-written, including a brief background on sexual and reproductive rights of youth, with a focus on Ethiopian youth, an overview of school-based and community-based interventions that have been implemented to address barriers to attaining such human rights, as well as the challenges encountered in the broader dissemination of such interventions. I applaud the authors for using a structured framework to guide their evaluation study, which was clearly defined and described (OH). All in all, the introduction was well-written. I would only recommend highlighting the study design in the purpose section of the manuscript.
  3. Materials and methods were also clearly delineated. Since this is a mixed methods approach, the authors used both quantitative and qualitative methods including thematic analysis for key informant interviews, appreciative inquiries, and desk reviews, and quantitative methods for descriptive statistics.  I would only recommend on elaborating on the type of descriptive statistics carried out in lines 166-168.
  4. The qualitative portion of the results was clearly presented with the thematic analysis portion clearly showing major themes and being supported by quotes from key informants. What is missing is the quantitative portion. I would recommend including a table to show your quantitative findings.
  5. The discussion supports the findings from the evaluation study with findings from the literature. I would recommend placing the strengths and limitations of your study before the conclusion towards the end of the discussion. I would also recommend highlighting other needed changes not only at the policy level but also at the community, household, and school levels for a more comprehensive approach to attaining such human rights. 
  6. The conclusion provides a good summary of the paper. No changes are needed.

Author Response

Comments and Suggestions for Authors

This is a well-written evaluation study using the OH framework to conduct a midterm evaluation of the Power to You(th) (PtY) programme in Ethiopia. Here are some minor comments for the authors:

  1. Please mention in the abstract that this is an evaluation study and which types of methods were used (qualitative/quantitative) to make it clear what type of study this is.

Thank you for  the feedback. In the abstract the feedback included.

  1. The introduction was well-written. I would only recommend highlighting the study design in the purpose section of the manuscript.

 Thank you for the feedback. We have included at the end of the introduction and beginning of the methodology to show clearly the study design and purpose.  

  1. Materials and methods were also clearly delineated. Since this is a mixed methods approach, the authors used both quantitative and qualitative methods including thematic analysis for key informant interviews, appreciative inquiries, and desk reviews, and quantitative methods for descriptive statistics.  I would only recommend on elaborating on the type of descriptive statistics carried out in lines 166-168.

Thank you very much for the feedback. We have included a detailed descriptives and what types of descriptives these included frequencies and percentages. These descriptive measures were not intended to establish causal impact but to complement the qualitative findings and illustrate the programme’s overall reach and contribution.

  1. The qualitative portion of the results was clearly presented with the thematic analysis portion clearly showing major themes and being supported by quotes from key informants. What is missing is the quantitative portion. I would recommend including a table to show your quantitative findings.

Thank you for the feedback. The results section is revised both your feedback and another feedback. Your feedback revised accordingly.

  1. The discussion supports the findings from the evaluation study with findings from the literature. I would recommend placing the strengths and limitations of your study before the conclusion towards the end of the discussion. I would also recommend highlighting other needed changes not only at the policy level but also at the community, household, and school levels for a more comprehensive approach to attaining such human rights. 

Thank you very much for the findings. The strengths and limitation are included just after the discussion.

Reviewer 2 Report

Comments and Suggestions for Authors

The manuscript addresses a critical area of public health in Ethiopia by evaluating a youth-led SRHR advocacy programme. Outcome Harvesting (OH) represents a cutting-edge and compelling methodological choice for capturing and mapping the complex, emergent changes from such an intervention. The paper has the opportunity to contribute significantly to the literature on both participatory evaluation methodologies and youth empowerment in SRHR. That said, the manuscript needs significant work prior to being considered for publication. The key issues are in the methodology not providing enough detail, and more importantly, the Results section has a very repetitive and confusing structure that diminishes the clarity and impact of your evidence. In ensuing comments, I will provide specific suggestions to assist you add rigor and relevance to your manuscript.

Introduction

  • Page 2, Lines 70-76: The introduction clearly sets-up the study and rationalizes the need for youth-focused SRHR interventions in Ethiopia. It also clearly articulates why the OH methodology should be used. To strengthen this point further, it could be helpful to provide a brief contrast of the OH approach to many conventional methodology/rationale (i.e., using predetermined, quantitative indicators, etc.) in the complex, adaptive programme you are working with "Power to You(th)". Thereby, further dissipating novelty for the reader.

Methods

  • Page 3, Lines 117-118: The manuscript indicates that "14 key outcomes were identified" after the prioritization process. Can you provide a description of the criteria – for example, were the outcomes selected based on the perceived importance of stakeholders, their relationship to the Theory of Change (TOC), their novelty, or some combination thereof?
  • Page 3, Lines 115-117: The manuscript states the team completed a "contribution analysis." Please provide a detailed explanation of how this was operationalized. In particular, how was the "level of contribution" of numerous actors defined, trialed, and verified throughout the workshop and the follow-up interviews? This is an important detail to understand the feasibility of the programme’s contribution to the realised outcomes.
  • Page 4, Lines 158-165: The discussion of the thematic analysis is somewhat vague, and I wonder if you could provide more detail regarding the coding. Were the codes deductive (based on the listed domains of the TOC you outline), or did you also invoke an inductive approach that would allow for the identification of unanticipated themes? This is important to clarify one of the strengths of OH is its ability to identify emergent, unintended outcomes.
  • Page 4, Lines 144-153: Although purposive sampling was used appropriately, explaining the specific number of participants involved in each of the interview categories (e.g., 30 substantiation interviews, 10 youth champions) would strengthen the overall methodological rigor of the study. 

Results

The key issue in the manuscript is the redundant structure of the Results section. Many of the same results (and in some cases, the exact same quotes) for the different findings are repeated under different headings/subheadings. For example:

  • The result of the reintroduction of Youth-Friendly Services at Shewa Robit is reported in Section 3.6 (Page 7, Lines 226-236), and again reported in Section 3.2 (Page 6, Lines 181-187).
  • The anecdote of the youth advocate preventing his sister's FGM on the other hand is reported in Section 3.3 (Page 6, Lines 191-195), and again in Section 3.7 (Page 7, Lines 242-247). The repetition of reporting the Results section occurs with every significant finding so the section becomes convoluted and the results lose impact.

I strongly suggest a full redesign of the Results section into an integrated and non-redundant narrative. Each thematic finding should only appear once. One logical way to present findings would be to organize them around the pathways outlined in the core TOC of your programme (see Methods for some examples (1) Promoting Youth Agency and Advocacy, (2) Changing Social Norms, (3) Institutional and Policy Change). This would give a much clearer and stronger presentation of your findings. Further, the authors note in the manuscript that complementary methods (e.g., Advocacy Maturation Tool), although this may be a minor point, the presentation of findings would be richer if you could explicitly draw on findings from those complementary tools into your results to show how those triangulated or added to the findings you presented in OH.

Discussion

  • Page 9, Lines 305-307, you mention a fascinating and complex unexpected outcome: "some youth resisted parent efforts to postpone marriage." This important finding is not picked up and discussed in the discussion. Please unpack this outcome, as it represents the non-linear pathways of social change that can arise and provides unique value of utilizing an OH methodology to capture this complexity.
  • Page 10, Lines 360-365: The limitations section is a solid beginning but should be more developed. Provide additional detail on the inherent limitations of the OH method, that is, the separation of the programme's contribution from full attribution of change and the possibility for selection bias (e.g., the danger of outcomes being given priority in favor of the most articulate or accessible participants). 

Author Response

Introduction

  1. Page 2, Lines 70-76: The introduction clearly sets-up the study and rationalizes the need for youth-focused SRHR interventions in Ethiopia. It also clearly articulates why the OH methodology should be used. To strengthen this point further, it could be helpful to provide a brief contrast of the OH approach to many conventional methodology/rationale (i.e., using predetermined, quantitative indicators, etc.) in the complex, adaptive programme you are working with "Power to You(th)". Thereby, further dissipating novelty for the reader.

Thank you very much for the feedback. We have revised accordingly in the main body of the manuscript.

Methods

  1. Page 3, Lines 117-118: The manuscript indicates that "14 key outcomes were identified" after the prioritization process. Can you provide a description of the criteria – for example, were the outcomes selected based on the perceived importance of stakeholders, their relationship to the Theory of Change (TOC), their novelty, or some combination thereof?

Thank you for your feedback. We have included the following in the main part of the research.

‘’A total of 14 key outcomes were prioritized through a participatory workshop with programme implementers and stakeholders. Outcomes were selected based on their relevance to the Theory of Change (TOC), meaning they directly reflected one or more pathways of change toward youth empowerment, gender norm transformation, and SRHR advocacy. In some cases, outcomes that were not explicitly anticipated in the TOC but that directly advanced the overarching programme goal were also included. Additionally, outcomes were prioritized when there was clear evidence of significant contribution by the programme and when the change was perceived as highly meaningful by stakeholders (e.g., influencing policy, shifting community norms, strengthening youth participation). This ensured that both expected and emergent, but substantively important, outcomes were captured and analyzed.”

  1. Page 3, Lines 115-117: The manuscript states the team completed a "contribution analysis." Please provide a detailed explanation of how this was operationalized. In particular, how was the "level of contribution" of numerous actors defined, trialed, and verified throughout the workshop and the follow-up interviews? This is an important detail to understand the feasibility of the programme’s contribution to the realised outcomes.

Thank you for the feedback. The following was provided in the main part of the study.

Contribution analysis was used to assess the plausibility of the programme’s role in each harvested outcome. For every outcome, the team first articulated a clear contribution claim that linked PtY activities to the observed change through specific mechanisms of influence. These claims were then explored in participatory workshops where implementers and stakeholders collectively mapped the broader “causal package.” This mapping process recognized that outcomes rarely result from a single cause, and therefore placed PtY activities—such as trainings, advocacy dialogues, and youth club strengthening—alongside the contributions of other actors including government initiatives, CSO partnerships, and contextual drivers. The focus was on identifying how programme actions helped to create an enabling environment for change or directly influenced actors, while acknowledging that the final realization of the change rested with those actors themselves. To test the plausibility of these claims, targeted evidence was gathered beyond the narrative outcome statements. Programme documents, meeting records, policy circulars, and service statistics were reviewed, while follow-up interviews with youth champions, community leaders, and government officials provided testimonies about the sequence and timing of changes. The credibility of the programme’s contribution was judged based on whether the observed changes followed logically after PtY interventions and whether external informants affirmed that these interventions had played a significant role, even if other factors were also at play. In this way, contribution was understood not as sole attribution of outcomes to the programme, but as a demonstration of how PtY inputs plausibly enabled or influenced the changes within its sphere of implementation.

  1. Page 4, Lines 158-165: The discussion of the thematic analysis is somewhat vague, and I wonder if you could provide more detail regarding the coding. Were the codes deductive (based on the listed domains of the TOC you outline), or did you also invoke an inductive approach that would allow for the identification of unanticipated themes? This is important to clarify one of the strengths of OH is its ability to identify emergent, unintended outcomes.

Thank you for the feedback. The following section was included .

The thematic analysis combined both inductive and deductive coding approaches to capture the full range of outcomes. In the first stage, an inductive approach was applied, whereby all harvested outcomes were coded without reference to a preset framework. This allowed themes to emerge directly from the data, preserving the bottom-up nature of Outcome Harvesting and ensuring that unanticipated or unintended outcomes were not overlooked. In the second stage, a deductive approach was applied to map these emergent themes onto the predefined domains of the programme’s Theory of Change (TOC), as well as donor reporting requirements and strategic priorities. This dual process ensured that the analysis remained faithful to the data while also facilitating alignment with the TOC, enabling the team to reflect on whether pathways remained realistic or required adjustment. In practice, this meant that inductively identified outcomes, such as unexpected youth-led advocacy efforts or the engagement of new religious leaders, were later situated within or alongside existing TOC pathways, ensuring that both emergent and anticipated outcomes were systematically captured.

  1. Page 4, Lines 144-153: Although purposive sampling was used appropriately, explaining the specific number of participants involved in each of the interview categories (e.g., 30 substantiation interviews, 10 youth champions) would strengthen the overall methodological rigor of the study. 

Thank you for the feedback.  We have included and addressed the feedback

In total, 51 interviews were conducted across different respondent categories. These included 30 substantiation interviews (25 from Amhara, 5 from Afar), 10 youth champions (8 from Amhara, 2 from Afar), and 11 key informants comprising CSO representatives, societal actors, and state officials (8 in Amhara, 3 in Afar). In addition, 12 programme implementers participated in the outcome harvesting workshop, and 13 stakeholders were engaged in the advocacy maturation assessment. This purposive sampling strategy ensured broad representation across youth, civil society, government, and community actors, thereby strengthening the methodological rigor and credibility of the findings

Results

  1. The key issue in the manuscript is the redundant structure of the Results section. Many of the same results (and in some cases, the exact same quotes) for the different findings are repeated under different headings/subheadings. For example: The result of the reintroduction of Youth-Friendly Services at Shewa Robit is reported in Section 3.6 (Page 7, Lines 226-236), and again reported in Section 3.2 (Page 6, Lines 181-187). The anecdote of the youth advocate preventing his sister's FGM on the other hand is reported in Section 3.3 (Page 6, Lines 191-195), and again in Section 3.7 (Page 7, Lines 242-247). The repetition of reporting the Results section occurs with every significant finding so the section becomes convoluted and the results lose impact. I strongly suggest a full redesign of the Results section into an integrated and non-redundant Each thematic finding should only appear once. One logical way to present findings would be to organize them around the pathways outlined in the core TOC of your programme (see Methods for some examples (1) Promoting Youth Agency and Advocacy, (2) Changing Social Norms, (3) Institutional and Policy Change). This would give a much clearer and stronger presentation of your findings. Further, the authors note in the manuscript that complementary methods (e.g., Advocacy Maturation Tool), although this may be a minor point, the presentation of findings would be richer if you could explicitly draw on findings from those complementary tools into your results to show how those triangulated or added to the findings you presented in OH.

Thank you for the feedback. The result section is filly revised and please refer to the revised  main  body of the manuscript

Discussion

  1. Page 9, Lines 305-307, you mention a fascinating and complex unexpected outcome: "some youth resisted parent efforts to postpone marriage." This important finding is not picked up and discussed in the discussion. Please unpack this outcome, as it represents the non-linear pathways of social change that can arise and provides unique value of utilizing an OH methodology to capture this complexity.

Thank you for the feedback. We have addressed and included the following in the main section of the manuscript.

One unexpected but significant outcome was that some youth resisted parental efforts to postpone marriage. While parental discouragement of early marriage is often celebrated, these cases reveal a more complex reality: some youth asserted their agency by rejecting parental control, even when framed as protective. This highlights the non-linear and sometimes paradoxical pathways of social change, where empowerment may not always align neatly with adult or community expectations. Outcome Harvesting was particularly well-suited to capture such nuanced and emergent dynamics, which would likely have been missed by conventional indicator-based evaluations. This underscores the value of OH in documenting the diversity of pathways through which change unfolds, especially in contexts where shifts in power relations between youth and adults are central to advocacy outcomes.

  1. Page 10, Lines 360-365: The limitations section is a solid beginning but should be more developed. Provide additional detail on the inherent limitations of the OH method, that is, the separation of the programme's contribution from full attribution of change and the possibility for selection bias (e.g., the danger of outcomes being given priority in favor of the most articulate or accessible participants). 

Thank you for the strength and limitation of the  study included in the main part of the manuscript

Round 2

Reviewer 2 Report

Comments and Suggestions for Authors

The authors have been diligent in addressing all the aforementioned critiques. The revisions are substantial, thoughtful, and have significantly improved the manuscript's scientific rigor, clarity, and overall quality.